# Transcriptome analyses in juvenile yellow perch (*Perca flavescens*) exposed *in vivo* to clothianidin and chlorantraniliprole: Possible sampling bias

Maeva Giraudo[1], Laurie Mercier[1], Andrée Gendron[1], Jim Sherry[2], Magali Houde[1]*

1 Environment and Climate Change Canada, Aquatic Contaminants Research Division, Montreal, Quebec, Canada, 2 Environment and Climate Change Canada, Aquatic Contaminants Research Division, Burlington, Ontario, Canada

* magali.houde@ec.gc.ca

**Data Availability Statement:** All relevant data are within the manuscript and its Supporting information files. Data will also be available via the

## Abstract

The St. Lawrence River is an important North American waterway that is subject to anthropogenic pressures including intensive urbanization, and agricultural development. Pesticides are widely used for agricultural activities in fields surrounding the yellow perch (*Perca flavescens*) habitat in Lake St. Pierre (Quebec, Canada), a fluvial lake of the river where the perch population has collapsed. Clothianidin and chlorantraniliprole were two of the most detected insecticides in surface waters near perch spawning areas. The objectives of the present study were to evaluate the transcriptional and biochemical effects of these two pesticides on juvenile yellow perch exposed for 28d to environmental doses of each compound alone and in a mixture under laboratory/aquaria conditions. Hepatic mRNA-sequencing revealed an effect of chlorantraniliprole alone (37 genes) and combined with clothianidin (251 genes), but no effects of clothianidin alone were observed in perch. Dysregulated genes were mostly related to circadian rhythms and to $Ca^{2+}$ signaling, the latter effect has been previously associated with chlorantraniliprole mode of action in insects. Moreover, chronic exposure to clothianidin increased the activity of acetylcholinesterase in the brain of exposed fish, suggesting a potential non-target effect of this insecticide. Further analyses of three clock genes by qRT-PCR suggested that part of the observed effects of chlorantraniliprole on the circadian gene regulation of juvenile perch could be the result of time-of-day of sacrifice. These results provide insight into biological effects of insecticides in juvenile perch and highlight the importance of considering the circadian rhythm in experimental design and results analyses.

## Introduction

The yellow perch (*Perca flavescens*; YP) is a widely distributed freshwater fish species indigenous to North America. In southern Canada, YP have a great importance for the local economy, cultural and recreational activities, especially in certain sectors of the St. Lawrence River (Quebec). Populations of YP have been abundant in the upstream sectors of the river but have

Environment and Climate Change Canada Open data catalogue.

**Funding:** This study was funded by Environment and Climate Change Canada and supports the St. Lawrence Action Plan.

**Competing interests:** The authors have declared that no competing interests exist.

suffered a severe decline in health and abundance downstream, particularly in Lake St. Pierre area where the population has collapsed at the beginning of the 2010s [1]. As a result, a moratorium on all commercial and sport fishing of YP in Lake St. Pierre was established in 2012 for an initial period of 5 years. However, due to the lack of significant population recovery, this moratorium was extended until 2027 [1, 2]. In this fluvial lake, continuous low levels of recruitment of YP age 1+ and 2+ have been observed since 2007 along with decreased growth rate of young-of-the-year resulting in lower overwinter survival of juveniles [3]. Low biomass and rapid sexual maturation in females have also been observed, which compromise the maintenance of a healthy reproductive stock [2].

A combination of different anthropogenic pressures have been identified as possible causing factors for the Lake St. Pierre population collapse, including overexploitation, increased predation by fishing birds, competition with invasive fish species, and changes in water temperature, level, and quality [4–6]. The loss of habitat caused by increasing agricultural activities has also been a major factor in the population decline, especially intensive annual cultures in floodplains, which are important spawning and nursery areas for YP of Lake St. Pierre [7]. The increased agriculture practices have also contributed to large inputs of high concentrations of nutrients and pesticides in Lake St. Pierre from tributaries draining surrounding farmlands, especially in the south watershed composed of more than 40% of agricultural lands [8, 9]. A recent survey of pesticide levels in Lake St. Pierre between 2008 and 2016 detected the presence of 12 to 21 different herbicides, insecticides and fungicides in surface water sampled from 19 sites across the Lake and in its tributaries, including at 11 sites located directly in YP spawning and nursery areas [10]. Among the insecticides, the neonicotinoid clothianidin and thiamethoxam were detected in all water samples at the maximum concentrations of 110 and 240 ng/L, respectively, which are 13 and 29 times above the chronic aquatic life criterion established in Québec for these substances (8.5 ng/L).

Neonicotinoids are systemic insecticides widely used around the world to protect crops against insect pests, primarily as seed coating for a variety of crops such as corn, potatoes, and soybean [11]. Their widespread usage has raised global concerns regarding their environmental fate as well as their impacts on non-target pollinator insects, especially bees [12, 13]. As a result, the use of the three neonicotinoids thiamethoxam, imidacloprid and clothianidin was prohibited in Europe as of December 2018 [13]. In Canada, these three substances have undergone risk assessment for pollinator insects by Health Canada's Pest Management Regulatory Agency's [14] and the risk of clothianidin and thiamethoxam to aquatic insects was reviewed [15, 16]. These assessments have led the Canadian Government to require mitigation measures on the use of thiamethoxam, imidacloprid and clothianidin, including restrictions on their usage as corn and soybean seed treatments. For example, the use of neonicotinoids or neonicotinoid-treated seeds is no longer permitted in the province of Quebec, unless an agronomic justification and prescription are obtained (https://www.environnement.gouv.qc.ca/pesticides/permis/modif-reglements2017/justification.htm). Accordingly, replacement products have been developed and are already used for crop protection. The diamide chlorantraniliprole is an example of a new class of chemical insecticide that has rapidly been replacing neonicotinoids and has been authorized in Canada for corn seed treatment [17, 18]. This compound was detected in 22% of water samples in Lake St. Pierre at maximum concentration of 13 ng/L [10] and more recent surveys in several of its tributaries showed that the concentrations peaks (220 ng/L) and frequencies of detection (100%) are on the rise [19]. Quality criteria have yet to be established for chlorantraniliprole as toxicological studies in non-target species are still very scarce. It was reported to have harmful effects on freshwater invertebrates at environmental concentrations 0.6–9.6 µg/L [20, 21], to alter ryanodine receptor activity in fathead minnows (*Pimephales promelas*) exposed for 96h to 0.025–10 µg/L [22], and to impair the

immune response of silver catfish (*Rhamdia quelen*) naturally infected with the bacteria *Aeromonas hydrophila* and exposed for 24h to 1.3 µg/L [23].

A number of *in situ* studies have assessed the health of Lake St. Pierre YP population at different levels of biological organization (from gene transcription and protein activities to physiological endpoints) comparing it to other populations from upstream sites. In adult YP, results showed lower body condition index, increased liver damages as well as the presence of oxidative stress and altered retinoid metabolism at the gene and protein levels, and lower activity of acetylcholine esterase (AChE) [24–27]. In juveniles and larvae, similar impacts on antioxidant and retinoid metabolism were observed as well as a potential interaction of UV radiations and neonicotinoids on the fish nervous system [28, 29]. However, all these studies were performed on fish sampled *in situ* where the combination of multiple anthropogenic and natural pressures made it impossible to identify the relative impact of each factor, in particular the potential effects of contaminants such as pesticides. The aim of the present study was to fill this gap by evaluating the chronic effects of two pesticides detected in the St. Lawrence River, clothianidin and chlorantraniliprole, in juvenile YP. Yellow perch from an aquafarm were exposed under laboratory conditions to environmental concentrations of the pesticides alone and in a mixture to assess the effects of these compounds on a most vulnerable stage of development. High-throughput sequencing of the transcriptome (RNA-sequencing) in liver was used to compare gene transcription profiles in YP and identify the modes of action of these pesticides; AChE analyses were also done in brain of fish as a neurotoxic biomarker.

## Materials and methods

### Plastic waste management

In compliance with the Ocean Plastics Charter [30], we adopted the 3-R approach (Reduce, Reuse, Recycle) for the management of scientific plastics. Whenever possible, single-use plastics were reused, or replaced either by reusable non-plastic products or by consumables made of recycled plastics. Non-hazardous plastic waste generated during this study was recycled locally by Groupe Gagnon (Prévost, QC, Canada); nitrile gloves were recycled by Kimberly-Clark (Huntsville, ON, Canada).

### Chemicals

Clothianidin (CAS 210880-92-5; 96% purity) and chlorantraniliprole (CAS 500008-45-7; 97% purity) were purchased from Toronto Research Chemicals (North York, ON, Canada). Triton X-100, 5,5-dithio-bis-(2-nitrobenzoic acid) (DTNB) and acetylthiocholine iodide were purchased from Sigma-Aldrich (Oakville, ON, Canada). Bradford reagent was purchased from Bio-Rad (Mississauga, ON, Canada).

### Yellow perch rearing and exposure

All animal procedures were approved by Environment and Climate Change Canada's Animal Care Committee working in accordance with the Canadian Council on Animal Care guidelines.

Newly hatched juvenile yellow perch were obtained from Trois Lacs fish culture station (Wolton, QC, Canada) and maintained in 400 L polyethylene tanks with flow-through municipal dechlorinated water and constant air bubbling at $15 \pm 2°C$ under a 16:8 h light: dark photoperiod for 3 months prior to exposure. Fish were fed daily with 1.5% (wet weight) of fish body weight of frozen adult brine shrimps (*Artemia salina*; Aquamerik, Levis, QC, Canada).

Fish (mean weight $0.92 \pm 0.1$ g) were randomly placed in four 30L glass aquariums (n = 37 fish/aquarium) and maintained in the same constant light, temperature, and air bubbling

conditions for a week-long acclimation period. Each aquarium was lined with a custom-made Vexar® LDPE (Masternet Ltd., Mississauga, ON, Canada) mesh basket that fit the size of the aquarium. These baskets allowed easy daily water renewal by transferring rapidly all fish in a new aquarium at once with minimum handling stress.

### *In vivo* exposure and chemical analysis

Each of the four aquarium was randomly assigned a treatment condition: solvent control (acetone; A), 200 ng/L clothianidin (CLO), 200 ng/L chlorantraniliprole (CH), and a mixture of 200 ng/L CLO and 200 ng/L CH (M). Treatment doses were chosen to be in the upper range of the concentrations of CLO and CH measured in surface water of the St. Lawrence River (Québec, Canada) and its tributaries [10]. Fish were exposed for 28 d to water spiked with concentrated stock solution of pesticides made in acetone to achieve the 200 ng/L nominal concentration. Vehicle control aquarium (0.03% acetone) was handled similarly.

Exposure concentrations were measured by the Centre d'expertise en analyse environnementale du Québec (CEAEQ) in Québec city (QC, Canada) to validate nominal doses. For each treatment condition, exposure media (40 ml, N = 3) was collected at the beginning of the experiment. Direct injection of 100 μl of exposure media was done on Oasis C18 HLB Direct Connect HP 20μm cartridges (Waters Corporation). Analyses were done by liquid chromatography coupled with a tandem quadrupole mass spectrometer (HPLC-MS/MS) equipped with an electrospray ionization source (ESI, TQ-S Waters Corporation). Water blanks were used during procedures and indicated no contamination. The limit of detection was 5 ng/L for clothianidin and 2 ng/L for chlorantraniliprole. Measured concentrations were within the 80–120% range of the nominal concentrations as required by the OECD test guidelines for testing of chemicals on juvenile fish (OECD 2000) (S1 Table).

Fish were fed daily throughout the exposure with 3% (wet weight) of fish body weight of frozen adult brine shrimps (*Artemia salina*; Aquamerik, Levis, QC, Canada). Spiked water was renewed daily and pH (7.8 ± 0.2), dissolved $O_2$ (92.5 ± 3.9%), and conductivity (297.7 ± 15.9 μs/cm) were monitored before and after water renewal. At the end of the 28d exposure, fish from each treatment aquarium were euthanized in 100 mg/L buffered MS-222 solution (ethyl 3-aminobenzoate methanesulfonate salt), measured (total length), and weighed (S2 Table). Liver tissue was sampled and stored at -80˚C in RNAlater™ stabilization solution (ThermoFisher Scientific, Mississauga, ON, Canada) for gene transcription analyses. Whole brain tissue was flash frozen at -80˚C for acetylcholinesterase (AchE) activity measurements. Remaining fish carcasses were stored at -80˚C for sex genotyping in muscle tissue. Tissue sampling of the 148 fish was performed over the course of the day, with 6h difference between control fish and the mixture treatment.

### Sex genotyping

Sex genotyping was carried out using PCR according to the protocol described in Feron et al. [31] with the following modifications (see S2 Table for results). Genomic DNA (gDNA) was extracted and purified from individual muscle tissue (≈25 mg) sampled from frozen yellow perch carcasses using the DNeasy® blood and tissue kit (Qiagen, ON, Canada) per the manufacturer's instructions. gDNA quantity and quality (A260/280>1.8; A260/230 > 2.0) were evaluated with a Nanodrop™ ND-2000 spectrophotometer (Thermo Fisher Scientific, ON, Canada). Primers were synthesized by IDT (Mississauga, ON, Canada): one primer pair was specific for the autosomal *amhr2a* gene present in both male and female perch (forward: 5′–GGGAAACGTGGGAAACTCAC–3′; reverse: 5′–AGCAGTAGTTACAGGGCACA–3′; product length: 638 bp), and one primer pair was specific for the Y chromosomal *amhr2by* gene

present only in males (forward: 5ʹ–TGGTGTGTGGCAGTGATACT–3ʹ; reverse: 5ʹ–ACTGTAGTTAGCGGGCACAT–3ʹ, product length: 443 bp). PCR were performed on a CFX96 Touch® realtime PCR detection system using HotSTarTaq Plus Master Mix kit (Qiagen, ON, Canada) with 50–200 ng of gDNA and a final concentration of 380 nM for each primer in a total reaction volume of 13 μL. The PCR conditions were as follows: 95˚C for 5 min, followed by 30 cycles of 94˚C for 30 s, 59˚C for 30 s and 72˚C for 1 min, and finishing by 10 min at 72˚C. PCR products were loaded on a 1.5% agarose gel in TBE 1X buffer, run at 100 V for 45 min and visualized using a ChemiDocTM Imaging system (Bio-Rad, ON, Canada).

## Library preparation and RNA-sequencing

For each experimental condition (A, CLO, CH, and M), total RNA was extracted from individual perch livers stored in RNAlater (n = 10 per treatment) using the RNeasy® plus mini kit (Qiagen, ON, Canada) per the manufacturer's instructions. RNA quality was assessed in an Agilent 2100 BioAnalyzer (Agilent, Santa Clara, CA, USA) using an RNA Nano Chip kit. RNA quantity was determined by analysis in a Qubit 3.0 Fluorometer (ThermoFisher, Waltham, MA. USA) using a Quant-iT RiboGreen® RNA assay kit (ThermoFisher, Waltham, MA, USA). The RNA concentration of each sample was then diluted to 8 ng/μL. mRNA libraries were prepared from 400 ng of each diluted RNA extract by means of the TruSeq®Stranded mRNA LT (Illumina, San Diego, CA, USA) which incorporated a two-stage sequential mRNA enrichment step. During library preparation, each library was labelled with a unique adapter index tag to permit multiplexed sequencing. The quality and fragment size distribution of each library was assessed in the Agilent 2100 BioAnalyzer using a DNA 7500 kit (Agilent, Santa Clara, CA, USA). Dimer free libraries were quantified in a Qubit 3.0 Fluorometer using the Quant-iT dsDNA Broad Range Assay kit and adjusted to 10 nM. The 39 resulting libraries (N = 9, 10, 10, 10 for control, CLO, CH, and M, respectively) were pooled by treatment (1.6 nM) and sequenced on a NextSeq™ 500 (Illumina, San Diego, CA, USA) using the Next-Seq®500/550 High Output kit v2 (150 cycles which provides 2 × 75 bp paired reads), resulting in a sequencing depth between 39–59 million reads per sample. Raw fastq files are available in the NCBI's Sequence Read Archive (SRA) under the accession number PRJNA742036.

## Reads alignment and differential transcription analysis

Processing of raw RNA-seq data was performed by the Canadian Center for Computational Genomics (Montreal, QC. Canada). Raw reads derived from the sequencing instrument were clipped for adapter sequence, trimmed for minimum quality (Q30) in 3' and filtered for minimum length of 32 bp using Trimmomatic [32]. Surviving read pairs were aligned to the yellow perch reference genome available at NCBI (PFLA 1.0; GCF_004354835) [31]. Genes from the corresponding.gff file were quantified using the GenPipes RNA-seq pipeline [33], which uses STAR and HTseq-count [34, 35]. Aligned RNA-Seq reads were assembled into transcripts and their relative abundance was estimated using Cufflinks [36] and Cuffdiff [37].

Exploratory analysis was conducted using various functions and packages from R and the Bioconductor project [38]. Differential transcription analysis was conducted using both edgeR [39] and Deseq [40]. For each gene, the effect of the different treatments (CLO, CH, M and the interaction) was tested for differential transcription with the limma-trend method [41]. Contrasts analyses were done to compare CLO+CH and M effects. Contrasts were specified using the corresponding linear model formula; B-A refers to the comparison of group B to A (control), using A as the denominator in the comparison. FC were therefore calculated by dividing the transcription values of each condition by the common denominator A: CLO/A, CH/A, M/A. For the interaction, the test verifies that exposure to the mixture differs from the sum of the

effects of the two pesticides taken individually: (M+A)/(CH+CLO). Thresholds for differential transcription were established using the adjusted *p*-value corrected by Benjamini-Hochberg false-discovery rate (FDR) of 0.05 and a fold change of ±1.5.

## Network interaction and functional enrichment analyses

The functional enrichment analysis was performed to further explore the biological functions and pathways associated with the significant differentially transcribed genes (DTGs). The list of DTGs identified in each condition (CH and M) with FDR < 0.05 were imported into Cytoscape v3.8.2 plug-in ClueGO [42, 43] to find significant enrichments using zebrafish (*Danio rerio*) as the reference organism. The selection criteria for Gene Ontology (GO) terms (Biological Process) and Kyoto Encyclopedia of Genes and Genomes (KEGG) pathways were based on two-sided hypergeometric tests with a *p*-value threshold < 0.05, which was corrected for multiple testing using the Benjamini-Hochberg FDR. Thresholds of a minimum of 3–8 GO levels, and a minimum number of 3 genes, or at least 4% genes in the respective terms, were applied.

Protein−protein interaction (PPI) networks were constructed to evaluate the relationship between DTGs based on the presence of gene fusions, neighboring genes, co-occurrence, experimental findings, text mining, database analysis, and evidence of co-expression using zebrafish as a reference organisms with a medium confidence score of 0.4 using the stringApp [44] in Cytoscape. Additional network clustering was performed to identify functional protein modules using Markov clustering (MCL) from the clusterMaker2 v1.3.1 [45] with an inflation value of 2, the String database score as an array source, and using an edge cut off of 0.4.

## qRT-PCR for gene transcription validation

Levels of mRNA of a subset of DTGs identified by RNA-seq in yellow perch liver were measured by quantitative real-time PCR (qRT-PCR) in independent liver samples (n = 12 per treatment) to validate RNA-seq results. Eight target genes were chosen among the differentially transcribed genes identified by RNAseq: six genes related to the circadian rhythm (*cry1*, *bmal1*, *rorb*, *clock*, *per1*, *cipc*) and two genes with the highest and lowest fold changes (*apoa4* and *gck*, respectively). Primer information is indicated in S3 Table and the procedure for the qRT-PCR analyses is detailed in S1 File. Percent efficiencies (90–110) and the coefficient of determination ($r^2 > 0.980$) of calibration curves respected the standards recommended in Taylor et al. [46].

To check for potential rhythmic variation in biological responses due to the time of sampling, mRNA levels of circadian gene orthologs were also measured by qRT-PCR in unexposed juvenile rainbow trout (*Oncorhynchus mykiss*) due to the lack of YP stock availability outside of the annual breeding period. Both fish species are teleosts and the controlled breeding of female rainbow trout in our laboratory allowed us to reproduce the exposure conditions of perch regarding size, weight, and carrying capacity. The sequencing of trout genome also facilitated the qRT-PCR analysis. Trout were maintained in the same conditions and sampled at the same time than for yellow perch (described in S1 File). Transcription levels of three circadian genes differentially transcribed in yellow perch exposed to pesticides (*cipc*, *lock*, and *per1*) were measured by qRT-PCR in trout liver (n = 10 per time point) (S3 Table and S1 File).

## Acetylcholinesterase activity

Yellow perch whole brains (n = 10 per condition) were homogenized using 5 mm stainless steel beads in a TissueLyser LT (Qiagen, ON, Canada) for 2 min (brain) at 50 Hz, in 10 mM Tris-HCl buffer (0.5 mL) with low salt Triton buffer (10 mM NaCl, 1% w/v Triton X-100, pH 7.3) [29]. Homogenates were centrifuged at 5000×*g* for 30 min at 4°C and the supernatant was used for triplicate measurement of AchE activity and total protein content.

Acetylcholinesterase activity was measured at 25°C in 0.1 M $NaH_2PO_4$ reaction buffer solution (pH 7.5) based on Ellman's method [47] and adapted to measure activities in 96-well microplates using 10 μL of supernatant for each sample, 0.4 mM of the chromogen DTNB, and 2 mM of the substrate acetylthiocholine iodide. Activity was monitored on a Synergy™ HT multi-detection microplate reader (BioTek, VT, USA) at 412 nm every minute for 15 min. Absorption kinetics were calculated in the linear range, then converted in nanomoles per minute according to the molar extinction coefficient of DTNB ($\varepsilon = 1.415 \times 10^4 \, M^{-1} \, cm^{-1}$). Results were normalized by the total protein concentration in supernatants measured by the Bio-Rad protein assay (Bio-Rad, Mississauga, ON, Canada) using the Coomassie Brilliant Blue G-250 dye (absorbance reading at 600 nm) with bovine serum albumin as a standard [48].

## Results and discussion

### Biological parameters

Sex determination could not be performed on juvenile individuals at the time of sequencing and was done *a posteriori* once a genotyping method had been published by Feron et al. [31]. Sex ratio was 50% males and females in all treatment conditions except for the control, which had 3 males and 7 females. Unsupervised exploratory analysis of the RNA-sequencing results revealed no effect of the sex of fish on the top 100 most variable genes (i.e., genes with the largest standard deviations of their $\log_2$ (CPM) values) (S1 and S2 Figs). Yellow perch reach sexual maturity at 2–3 years for males and 3–4 years for females [5, 49]. A previous study characterizing the sex-related genes by RNA-sequencing in 1+ year-old yellow perch reared in experimental tanks has reported no sex-biased gene expression in muscle of female and only a small number of genes [55] were found to be specifically expressed in males [50]. Based on these observations and the lack of significant effect of sex in the present analysis, the effect of pesticide treatments was analyzed without the sex factor for subsequent transcriptome profiling. Weight and length of fish were not significantly different among treatment groups (weight: F = 0.1327; P = 0.9404; length: F = 0.9391; P = 0.4236).

### Yellow perch hepatic transcriptional response

The summary of data generated by RNA-sequencing of yellow perch liver are reported in S4 Table. The overall alignment rate of the reads to the yellow perch reference genome ranged from 94.8 to 97.5% per sample, resulting in a total of 28,903 genes detected. Because very low counts across all samples provide little evidence for differential expression and because they add to the multiple testing burden when estimating false discovery rates, lowly expressed genes that did not reach a minimum of 1 count per millions reads (CPM) in at least ten samples, as calculated with edgeR, were filtered out [51]. This CPM value corresponds to a minimum of 10 counts in our libraries, which is the recommended threshold. Following this procedure, 13, 245 filtered genes were retained for further analysis.

A cluster of 43 of the most variable genes seemed to be co-regulated in some individuals, independently from the treatment condition (S1 Fig). These genes coded for digestive enzymes (e.g., trypsins, chymotrypsins, elastases, lipases and carboxypeptidases) and for yolk-related proteins (e.g., choriolytic enzymes and vitellin membrane proteins). The activity of these proteins is directly related to the ontogenic development of the digestive tract of fish larvae, showing variation depending on the larval stage [52]. In YP, the larval stage extends from early spring to the end of summer/beginning of fall, when the first juvenile stage appears [53]. In the present study, newly hatched YP larvae were acclimated for a 3-months period before being used for experimentation in September, which would correspond to the first juvenile stage. However, fish were maintained at a lower temperature (15°C) than the water temperature in

the natural YP habitat at the same period, slowing down the developmental process. It is therefore possible that fish were still undergoing larval ontogeny during the exposure period, which could explain the clustering of the most variable genes observed in the heatmap (S1 Fig). Further differential analysis of gene transcription was used to identify more specifically the potential effect of the pesticide exposure.

## Differential analysis of gene transcription

Differentially transcribed genes (DTGs) were identified in YP by comparing the levels of transcripts obtained by RNA-sequencing in each treatment condition (CH, CLO, and M) against unexposed fish (A). The interaction of CLO and CH was also tested to see how much the effect of exposure to the mixture (M) differed from the summed effect of both pesticides taken individually (CLO+CH). The list of all DTGs identified after Benjamini-Hochberg correction ($p$-adj $< 0.05$) are shown in S5 Table. Results from RNAseq measurements were validated by qRT-PCR analyses on a suite of 9 genes both for the direction and the magnitude of the regulation (Pearson correlation coefficients $R^2 > 0.9$). Validation results are shown in S5 Fig.

The neonicotinoid clothianidin did not affect the regulation of gene transcription in the liver of YP exposed for 28 d at environmental concentration (Fig 1A). Despite the large body of literature on the effects of neonicotinoid insecticides on non-target invertebrate and mammals, the

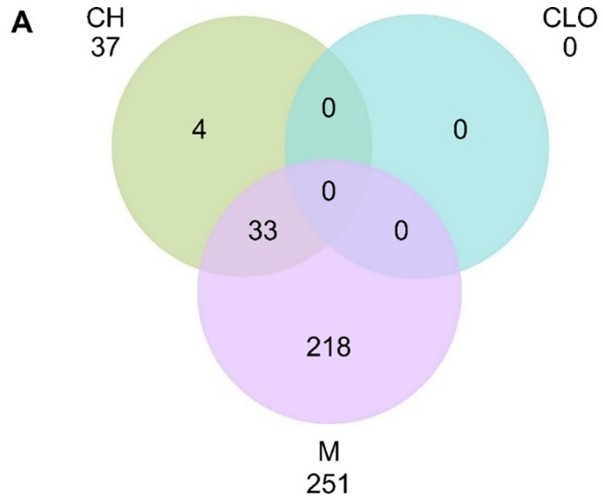

| Contrast | Nb. DTGs up-regulated | Nb. DTGs down-regulated |
|---|---|---|
| CLO-control | 0 | 0 |
| CH-control | 26 | 11 |
| M-control | 76 | 175 |
| Interaction (M+control)-(CH+CLO) | 0 | 0 |

**Fig 1. Differentially transcribed genes (DTGs) in liver of yellow perch exposed for 28 d to clothianidin (CLO), chlorantraniliprole (CH) and the mixture of both (M).** Only genes with a fold-change of 1.5 and an adjusted $p$-value $< 0.05$ were considered differentially transcribed. (A) Intersection of DTGs in each condition. (B) Number of up- and down-regulated DTGs.

chronic, sub-lethal effects of clothianidin in fish is still largely unknown. A recent study showed no effect of clothianidin on anti-predatory behavior in zebrafish larvae exposed for 24h at environmental concentration (3μg/L) [54]. Another study in larvae of sockeye salmon (*Oncorhynchus nerka*) reported no significant effects of 118 d exposure to environmentally relevant concentration of clothianidin (0.15–150 μg/L) on survival, hatching, growth or deformities, and found an increase in whole body 17β-estradiol levels at 0.15μg/L [55]. This study was the first to look at the transcriptional effect of clothianidin in fish and observed the dysregulation of only one gene out of the five measured in the liver (*glucocorticoid receptor 2*) at the highest concentration (150 μg/L). Altogether, the published information and the results from the present study suggest that concentrations of clothianidin that juvenile YP might be exposed to in the wild do not cause detectable sublethal effects at the transcriptional level in the liver after 28 d.

Chlorantraniliprole affected the transcription of 37 genes in the liver of YP, most of which were up-regulated (Fig 1B). Most of CH-dysregulated genes were also found differentially transcribed in the same direction by the mixture of CH and CLO (33 genes in common) and the transcription of 218 DTGs was dysregulated by an exposure to the mixture but not to individual pesticides. However, the effect of the mixture was not significantly different compared to the summed effect of CLO+CH (Fig 1B). This could indicate an absence of synergistic or additive effect of CH and CLO on the transcriptional response of YP at the concentration studied. Synergistic interactions between pesticides are seldom reported in environmental toxicology studies and mostly involve compounds with similar modes of action [56, 57]. Clothianidin and chlorantraniliprole have two distinct modes of action in insects: clothianidin is a neurotoxic agent acting as an agonist of the nicotinic acetylcholine receptors (nAChRs), while CH acts by activation of the ryanodine receptors in insect nerve and muscle cells. However, the mode of action of these two substances and their potential interactions in fish are unknown. Further functional analyses of the genes and biological pathways affected in yellow perch exposed to CLO and CH was performed to understand the increased transcriptional dysregulation observed in YP in response to the mixture of both pesticides.

## Functional analyses of differentially transcribed genes

Pathway and network analyses of DTGs in the liver of YP exposed to CH resulted in significant enrichment of pathways involved in circadian regulation of gene expression and photoperiodism (Fig 2 and S6 Table). Two of the genes associated with these enriched pathways were also the most significantly upregulated genes in response to CH: *cryptochrome-1* (*cry1*) and *aryl hydrocarbon receptor nuclear translocator-like protein 1* (*arntl*) (Fig 3). The three most significantly downregulated genes *hepatic leukemia factor* (*hlf*), *CLOCK-interacting pacemaker* (*cipc*), and *thyrotroph embryonic factor* (*tef*) were also found in a cluster of genes related to circadian rhythms along with 9 genes differentially transcribed in response to CH as identified through PPI network analysis (Fig 4). Differentially transcribed genes identified in response to M were associated with 16 significantly enriched GO and KEGG pathways, among which the circadian regulation of gene expression was also the most prominent pathway with the largest percentage of associated genes (Fig 2 and S7 Table). The circadian genes *cipc*, *hlf*, *arntl* and *cry1* were among the most dysregulated genes in response to M, with an increased significance compared to CH (Fig 3). These genes were also found in the cluster of 15 interacting genes related to circadian rhythms, with 9 genes found in both CH and M circadian clusters (Fig 4).

The circadian system is a natural biological process that synchronizes endogenous rhythms with environmental factors such as light/day and feeding/fasting cycles in all animals, including fish [58, 59]. These daily circadian rhythms are controlled by internal molecular clocks formed by positive and negative transcriptional feedback loops that involves the transcription

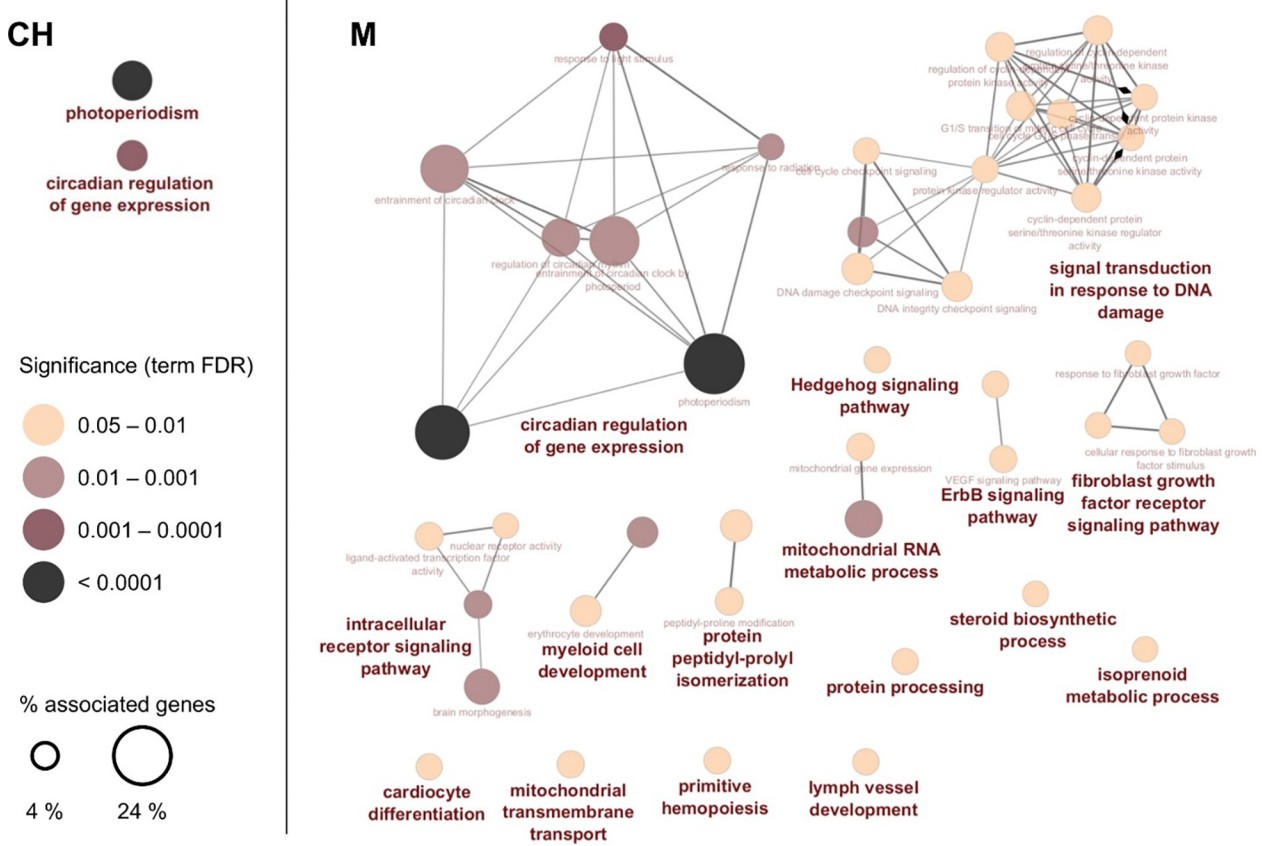

**Fig 2. Functionally grouped gene network analysis for yellow perch exposed to chlorantraniliprole (CH) and a mixture of CH and clothianidin (M).** GO-terms and KEGG pathways were obtained through a functional enrichment analysis of the DTGs using the ClueGO plugin in Cytoscape. Each node represents a significantly enriched GO or KEGG term (hypergeometric test $p < 0.05$ with Benjamini-Hochberg correction). The node color corresponds to the significance of the enriched pathway and the node circle diameter to the percentage of associated genes for that pathway. The most significant enriched terms of each functional group are illustrated as a summary label in bold.

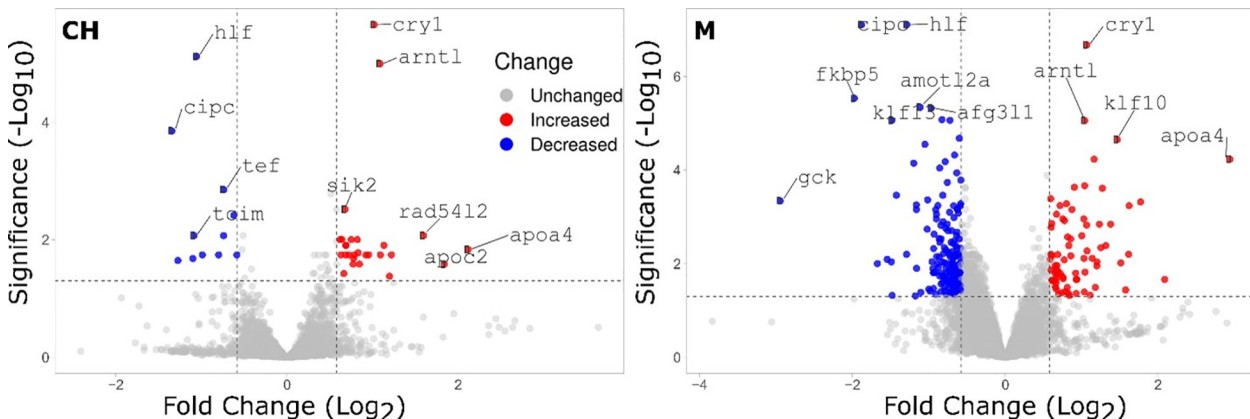

**Fig 3. Volcano plots depicting the log2 fold changes for gene transcription levels in response to chlorantraniliprole (CH) and the mixture of CH and clothianidin (M) in yellow perch liver.** Genes highlighted in red and blue are significantly up- and down-regulated, respectively (log$_2$ fold change $\pm1.5$, adjusted *p*-value <0.05).

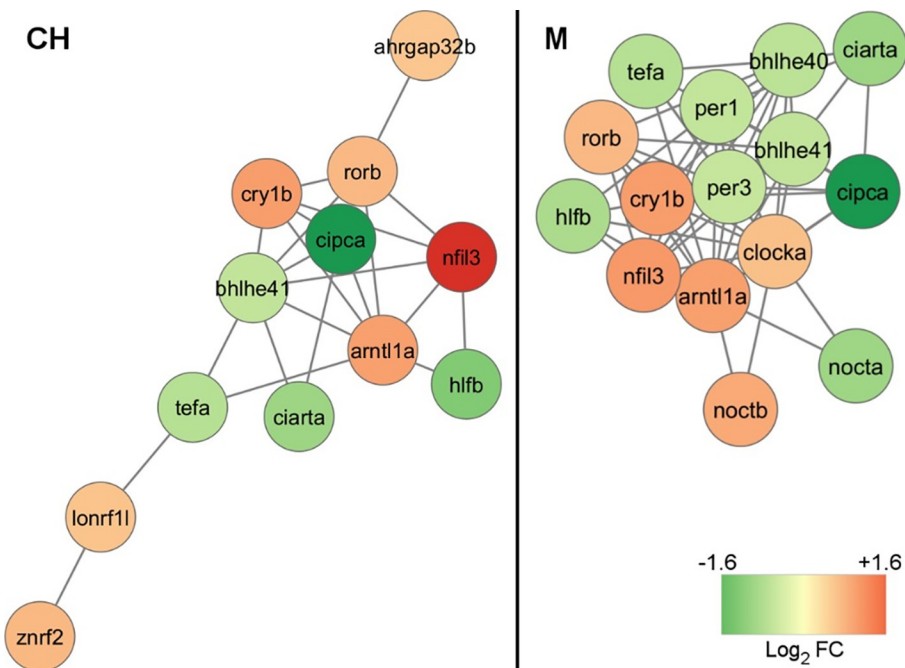

**Fig 4. Functional genes modules associated with circadian rhythms obtained after MCL clustering of the PPI network using the STRING App in Cytoscape (S3 and S4 Figs).** Red and green nodes represent upregulated and downregulated genes, respectively, in yellow perch exposed to chlorantraniliprole (CH) and a mixture of CH and chlothianidin (M).

factors *circadian locomotor cycles kaput* (*clock*), *arntl* and the genes from the *period circadian protein* (*per*) and *cry* [60–62]. These core clock genes create daily oscillations in the transcription of a multitude of target genes, which in turn affect different cell functions as well as many physiological, metabolic and behavioral parameters [63]. Transcriptional disruption of circadian rhythm can therefore have negative impacts on organismal biology and potential ecological consequences. Several evidence have shown the potential of environmental contaminants to disrupt circadian rhythms in fish [64–66]. Chronic exposure (21 d) to the neonicotinoid imidacloprid (10 µg/L) was notably found to disrupt the 24-hour expression rhythm of important clock genes (*clock*, *per and cry*) in zebrafish (*Danio rerio*), a molecular effect associated with behavioral disorders in the exposed fish [67]. In the present study, the transcription of the four core clock genes (*clock*, *arntl*, *cry1* and *per*) as well as 17 clock-controlled genes was significantly dysregulated by CH alone and in combination with CLO (Fig 4 and S6 and S7 Tables), suggesting a potential chronic disruption of circadian cycles in juvenile perch by CH.

In this study, we also observed that genes coding for most of the Ca2+ signaling proteins (i.e. camkk1, casq1, kdm6b, and prkca) were downregulated by exposure to M (S7 Table). Chlorantraniliprole and other diamide insecticides have been designed to target insect ryanodine receptors (RyRs) by promoting excessive intracellular Ca2+ release, causing impaired muscle contraction, feeding inhibition, and ultimately leading to insect paralysis and death [67]. Although diamides have 500 times higher selectivity for the insect RyR over the mammalian receptor [67], it was recently reported that CH was able to bind to the RyR of the fish *Pimephales promelas* [22], causing an approximate 500% maximal response comparable to the one from the invertebrate *Daphnia magna*. Authors also found changes in transcription levels of the genes coding for two RyRs and for the Ca2+ pump SERCA1 in *P. promelas* exposed for

96-h to 25–10000 ng/L of CH. Altogether, these recent reports and the results from the present study suggest that CH can impact cellular Ca2+ homeostasis and signaling in fish.

In mammals and Drosophila, intracellular Ca2+ signaling has been shown to coordinate the molecular rhythms of clock genes through a variety of mechanisms including RyR, Ca2+-binding proteins such as calsequestrins (CASQ), and phosphorylation of clock proteins (e.g. arntl, clock and lysine demethylase) by protein kinases such as PKC and Ca2+/calmodulin-dependent protein kinase (CaMK) [68, 69]. This could suggest that the dysregulation of circadian genes observed in YP might be caused by the activation of RyR by CH and by the resulting Ca2+ signaling dysregulation.

Knowing that feeding schedules are the main regulating factors (i.e., zeitgebers) of the clock gene oscillations in the fish liver [70, 71], it is also possible that the dysregulation of clock genes transcription measured in YP exposed to CH and M was partly induced by the morning feeding schedule maintained throughout the exposure period. Indeed, sampling was performed over a 6h period in the following order: A, CLO, CH, and M, which corresponded to early morning, late morning, early afternoon, and late afternoon, respectively. Measurements by qRT-PCR in rainbow trout showed similar oscillation of the transcription levels of three circadian genes (*cipc*, *clock*, *and per1*) according to the sampling time as in YP as shown in S6 Fig. At this stage, it is therefore difficult to discern the potential transcriptional effect of CH from that of the time of sampling on circadian gene regulation. In animals, the core molecular clock machinery directly or indirectly controls the regulation of about 10 to 20% of the expressed genome [69, 72]. Thus, part of the transcriptional response observed in response to CH and M might be the result of such circadian regulation, independently of the treatment. The present results underline the importance of accounting for the potential influence of circadian rhythms on experimental design, sampling and result interpretation [59]. Acknowledging both the influence of the circadian rhythm on the response of organisms to contaminants and the effect of contaminants on circadian rhythm regulation, future studies should include a mean of measuring the temporal factor. As such, Prokkola and Nikinmaa [61] advocated that from now on, at least three time points in a daily cycle should be considered in experimental design instead of a single one (for all treatment groups including control).

## Acetylcholinesterase activities in YP

Measurement of AChE activity in brain of YP showed significantly increased levels in individuals exposed to CLO alone and in a mixture with CH (Fig 5). Increased AChE enzyme activities have also been observed in YP larvae exposed for 7 d to 8.33 and 23.32 ng/L of another neonicotinoid: the pesticide imidacloprid [29]. Similarly, AChE activity was also found to increase in the brain of juvenile Chinese rare minnows (*Gobiocypris rarus*) exposed to 0.5 and 2.0 mg/L of imidacloprid during 60 d [68]. AChE is an important enzyme that plays a role in degrading the neurotransmitter acetylcholine and it can be a target for pesticides. Neonicotinoids do not target AChE directly but rather act as agonists of the nicotinic acetylcholine receptors (nAChRs) of insects by binding selectively to the receptors in place of acetylcholine [69]. This binding prevents the inactivation of the receptors by AChE, which results in insects in the overstimulation of the nervous system and ultimately its death. Although neonicotinoids have a stronger specificity and affinity for insect nAChRs than for vertebrate receptors [11, 70], the modes of action might be conserved in fish. Therefore, CLO could have had similar impacts on nAChRs in YP brains resulting in accumulation of acetylcholine at the cholinergic synaptic clefts. Hence, the induction of AChE activity could–be a compensatory response mechanism to degrade excess acetylcholine, as has been suggested in zebrafish exposed to 0.87–3.51 mg/L of sulfoxaflor for 24 to 96 h [71].

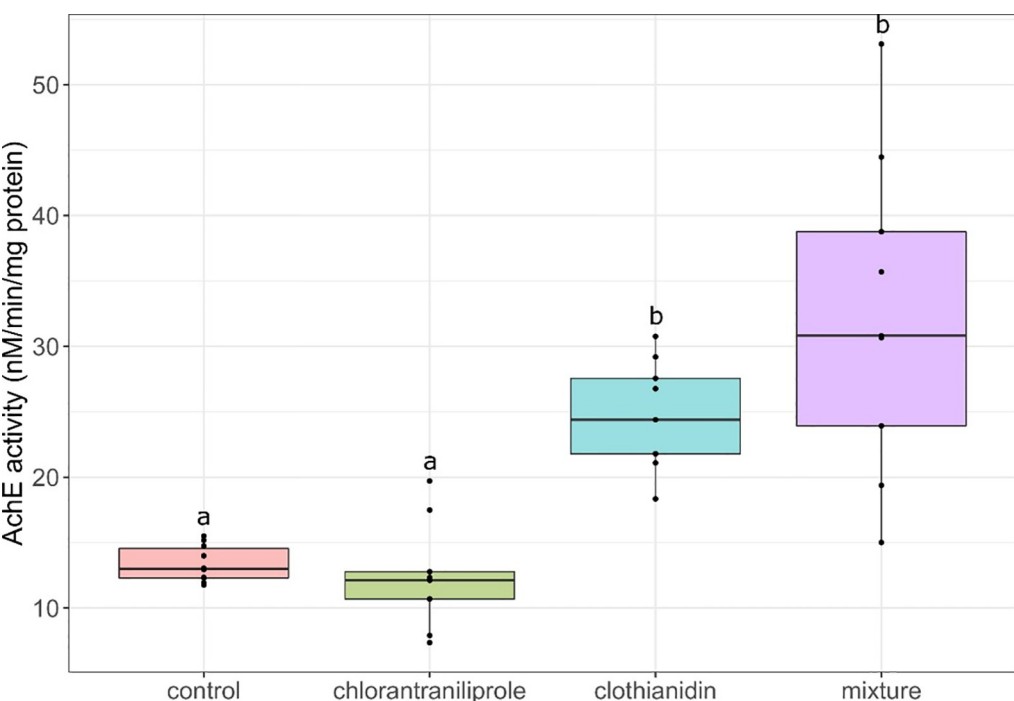

**Fig 5. Boxplot of acetylcholinesterase (AchE) activity measured in brain tissue from yellow perch exposed to pesticides and represented as minimum, maximum and median values, with 25% and 75% quantiles.** Different letters indicate significant differences between treatment conditions ($p < 0.001$).

## Conclusions

This study investigated the effects of two pesticides found in natural yellow perch habitats, CLO and CH, on gene transcription levels in juvenile perch exposed in laboratory-controlled conditions to the environmental doses of the compounds alone and in a mixture. Results suggested that only CH affected the gene transcription that were mostly related to circadian rhythms, which could be a result of the pesticide itself and/or the time-of-day of sacrifice. This observation points out the importance of circadian rhythm in experimental design and results analyses. Chlorantraniliprole also affected the transcription of genes involved in $Ca^{2+}$ signaling and homeostasis, which is associated with the mode of action of CH in insects; this could have indicated similar molecular impacts in fish for this compound. Clothianidin exposure increased the activity of acetylcholinesterase in the brain of YP, suggesting similar targets in fish than in insects. The present results provide valuable insight into non-target biological effects of insecticides in juvenile perch at low environmental doses. Further analyses are necessary to better understand the effects of these pesticides on fish in impacted environments, especially regarding the frequency and rate of pesticide use. In natural environments, fish are not exposed to constant, linear concentrations of pesticides, but rather to "pulses" of sometimes very high concentrations of pesticides resulting from seasonal culture treatments and soil drainage by the rain. The biological effects of such pulse exposures remain to be elucidated for juvenile YP from the Lake St. Pierre.

## Supporting information

**S1 File. Rainbow trout maintenance and sampling and quantitative real-time PCR (qRT-PCR).**
(DOCX)

**S1 Fig. Heatmap of the top 100 most variable genes in juvenile yellow perch liver exposed to pesticides (A: Acetone solvent control, CH: Chlorantraniliprole, CLO: Clothianidin, M: Mixture of both pesticides).** Expression values of samples for genes with > 1 CPM in at least 10 samples were ordinated using hierarchical clustering using only the top 100 genes with the largest standard deviations of their log2(CPM) values.
(DOCX)

**S2 Fig. Multidimensional scaling (MDS) plot of the filtered genes detected by RNA-seq in yellow perch liver samples (n = 37) exposed to pesticides (A: Control, CH: Chlorantraniliprole, CLO: Clothianidin, M: Mixture of both pesticides).**
(DOCX)

**S3 Fig. Protein-protein interaction network of DTGs in the liver of juvenile YP exposed to chlorantraniliprole using the STRING App in Cytoscape.** Red and green nodes represent upregulated and downregulated genes, respectively. Zebrafish orthologs of each perch DTG were used as gene list input.
(DOCX)

**S4 Fig. Protein-protein interaction network of DTGs in the liver of juvenile YP exposed to a mixture of clothianidin and chlorantraniliprole using the STRING App in Cytoscape.** The red and green nodes represent upregulated and downregulated genes, respectively. Zebrafish orthologs of each perch DTG were used as gene list input.
(DOCX)

**S5 Fig. Correlation between the relative quantification of gene transcription levels measured by qRT-PCR and RNAseq in yellow perch exposed to chlorantraniliprole (CH, blue) and a mixture of CH and clothianidin (M, red).** Data are expressed as the $\text{Log}_2$(fold change) of the mean transcription values between treatment and control conditions (n = 12 for qRT-PCR and N = 10 for RNAseq).
(DOCX)

**S6 Fig.** Relative gene transcription levels of circadian genes measured by qRT-PCR in yellow perch exposed to pesticides (A) and in rainbow trout sampled at different times of day (B). Data are expressed as the fold change of the mean relative transcription values (n = 12 for perch, n = 10 for trout). Asterisks indicate a significant difference from the control (A) or from early am sampling time (B).
(DOCX)

**S1 Table. Nominal and measured concentrations of clothianidin and chlorantraniliprole in water from exposure aquarium (mean of N = 3 ± SD).** Samples were collected at the beginning of the exposure.
(XLSX)

**S2 Table. Biological information of yellow perch from the different treatment.**
(XLSX)

**S3 Table. Primers used for qRT-PCR analyses in yellow perch and rainbow trout.**
(XLSX)

**S4 Table. Summary of sequencing data generated by RNA-seq for yellow perch exposed to pesticides.**
(XLSX)

**S5 Table. List of all differentially transcribed genes measured by RNA-sequencing in liver of yellow perch after 28d exposure to chlorantraniliprole (CH) and a mixture of clothianidin and CH (M).** Genes significantly up- and down-regulated (adjusted p-value<0.05, FC ±1.5) are indicated in red.
(XLSX)

**S6 Table. Statistical enrichment analysis of over-represented Gene Ontology (GO) terms associated with differentially transcribed genes in liver of juvenile yellow perch exposed to chlorantraniliprole for 28 d.** Zebrafish orthologs were used as gene list input in the Cluego plugin in Cytoscape.
(XLSX)

**S7 Table. Statistical enrichment analysis of over-represented Gene Ontology (GO) and KEGG terms associated with differentially transcribed genes in liver of juvenile yellow perch exposed to a mixture of clothianidin and chlorantraniliprole for 28 d.** Zebrafish orthologs were used as gene list input in the ClueGO plugin in Cytoscape.
(XLSX)

## Acknowledgments

The authors would like to acknowledge Émilie Lacaze, Maude Lachapelle, Rebecca Gouge, Labrini Vlassopoulos, Valérie Grenon, Jessy Côté, and Michel Defo for their assistance during fish sampling. We also thank Tannis Neheli for assistance with RNAseq.

## Author Contributions

**Conceptualization:** Maeva Giraudo, Andrée Gendron, Magali Houde.

**Data curation:** Maeva Giraudo, Jim Sherry.

**Formal analysis:** Maeva Giraudo, Laurie Mercier, Jim Sherry.

**Funding acquisition:** Magali Houde.

**Investigation:** Andrée Gendron.

**Methodology:** Maeva Giraudo, Laurie Mercier, Jim Sherry.

**Supervision:** Magali Houde.

**Validation:** Andrée Gendron, Jim Sherry.

**Writing – original draft:** Maeva Giraudo, Laurie Mercier, Andrée Gendron.

**Writing – review & editing:** Maeva Giraudo, Laurie Mercier, Andrée Gendron, Magali Houde.

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
