## [Decision Letter · Decision Letter 0]

19 Dec 2023

PONE-D-23-30911Transcriptome analyses in juvenile yellow perch (Perca flavescens) exposed in vivo to clothianidin and chlorantraniliprole: possible sampling biasPLOS ONE

Dear Dr. Houde,

Thank you for submitting your manuscript to PLOS ONE. After careful consideration, we feel that it has merit but does not fully meet PLOS ONE’s publication criteria as it currently stands. Therefore, we invite you to submit a revised version of the manuscript that addresses the points raised during the review process.

Please see the comments from two reviewers below, both of whom have raised significant concerns related to the study design. We now invite you to address these.Please note that PLOS ONE has policies to allow publication of all valid research, including null and negative results.

We look forward to receiving your revised manuscript.

Kind regards,

Hanna Landenmark

Staff Editor

PLOS ONE

“The authors would like to acknowledge Émilie Lacaze, Maude Lachapelle, Rebecca Gouge, Labrini Vlassopoulos, Valérie Grenon, Jessy Côté, and Michel Defo for their assistance during fish sampling. We also thank Tannis Neheli for assistance with RNAseq. This study was funded by Environment and Climate Change Canada and supports the St. Lawrence Action Plan.”

“This study was funded by Environment and Climate Change Canada and supports the St. Lawrence Action Plan.”

Reviewers' comments:

Reviewer's Responses to Questions

**Comments to the Author**

1. Is the manuscript technically sound, and do the data support the conclusions?

Reviewer #1: Yes

Reviewer #2: Partly

2. Has the statistical analysis been performed appropriately and rigorously? 

Reviewer #1: Yes

Reviewer #2: Yes

3. Have the authors made all data underlying the findings in their manuscript fully available?

Reviewer #1: Yes

Reviewer #2: Yes

4. Is the manuscript presented in an intelligible fashion and written in standard English?

Reviewer #1: Yes

Reviewer #2: Yes

5. Review Comments to the Author

Reviewer #1: This manuscript reports a study investigating the effect of clothianidin and chlorantraniliprole on the hepatic gene expression of juvenile yellow perch exposed to each compound alone and in a mixture for 28 days. Novel information regarding circadian rhythm effects in experimental design and results analyses was suggested. Overall, the experiments appear to be robustly performed.

Since the changes in gene expression have been analyzed after exposure for 28 days, therefore, it is likely that these enriched genes may be responsible for the effects related to compensation. Potentially supporting my argument is that the data presented from relative expression levels of circadian genes in Figure S6, and the similar weight and length across all groups (table S2).

Reviewer #2: The study “Transcriptome analyses in juvenile yellow perch (Perca flavescens) exposed in vivo to clothianidin and chlorantraniliprole: possible sampling bias” examined the effects of two insecticides on their own and in combination on juvenile yellow perch liver transcriptomes and brain acetylcholinesterase activity. I would first like to commend the authors on the extent of work that went into the study, and their ability to communicate their findings in a comprehensive and succinct manner.

However, despite finding potential impacts of insecticide impacts on genes associated with calcium signalling, and changes in activity of acetylcholinesterase activity in the brain, a large extent of their significant results appear to be due to differences in sampling times for their treatment groups. I appreciate the effort the authors put into verifying potential impacts of sampling time by examining similar clock genes in rainbow trout, but it does unfortunately highlight an important error in the design of the experiment. If the authors wish to publish the study in its current form, I believe that discussion of the differentially transcribed clock genes needs to be greatly reduced. The connection the authors make between calcium signaling genes and clock genes (page 20) also raises questions regarding whether transcription of calcium signaling genes could also be sensitive to the sampling time. Considering the arguments made by the authors on lines 469-470 – “At this stage, it is therefore impossible to conclude on the transcriptional effect of CH on circadian gene regulation” and that they emphasize that these genes regulate 10-20% of the expressed genome (lines 471-472), it becomes difficult to discern what this study offers with respect to our knowledge on the effects of the studied insecticides on the yellow perch liver transcriptome.

Minor comments:

Lines 269-276 – can the authors provide further justification as to why juvenile rainbow trout were an acceptable alternative to yellow perch? This point may be mute, given the end result.

Line 307-308 – were differences in weight and length among treatment groups analyzed statistically?

Line 314 – Can the authors include their reasoning on why a minimum of 1 CPM in at least ten samples was chosen as the threshold for DTG filtering.

Can the authors elaborate on how the comparison was made between the CLO+CH and M effects on differentially transcribed genes.

6. PLOS authors have the option to publish the peer review history of their article (what does this mean?). If published, this will include your full peer review and any attached files.

Reviewer #1: No

Reviewer #2: No

---

## [Author Response · Author response to Decision Letter 0]

26 Jan 2024

Responses to Reviewers and Publisher

Thank you for the review of our manuscript PONE-D-23-30911 - Transcriptome analyses in juvenile yellow perch (Perca flavescens) exposed in vivo to clothianidin and chlorantraniliprole: possible sampling bias. We appreciate the thoughtful comments provided by the two reviewers. Our responses can be found below. Page numbers indicated refer to the track-change revised manuscript.

As part of the revised manuscript submission we have included a track-changes version of the original paper showing all changes made in response to the reviewers comments. 

Responses to reviewers -

Reviewer #1: This manuscript reports a study investigating the effect of clothianidin and chlorantraniliprole on the hepatic gene expression of juvenile yellow perch exposed to each compound alone and in a mixture for 28 days. Novel information regarding circadian rhythm effects in experimental design and results analyses was suggested. Overall, the experiments appear to be robustly performed.

Since the changes in gene expression have been analyzed after exposure for 28 days, therefore, it is likely that these enriched genes may be responsible for the effects related to compensation. Potentially supporting my argument is that the data presented from relative expression levels of circadian genes in Figure S6, and the similar weight and length across all groups (table S2). 

Response: Although possible, it would be highly speculative to suggest that the observed lack of difference in growth between pesticide-exposed and control fish is the manifestation of a compensatory response to insecticide exposure. This could for instance imply that exposed perch have ingested more food than control fish to compensate for energy loss due to pesticide exposure. Unfortunately, we have no data to support such hypothesis. However, the increase in AChE activity observed after exposure to clothianidin alone and in mixture could indeed be a secondary response of the neurons to the insecticide. Clothianidin is not known to directly influence the activity of AChE; rather, it binds selectively to nicotinic acetylcholine receptors. This binding prevents the inactivation of the receptors by AChE, which results in an overstimulation of the nervous system. In this case, the observed increase in AChE activity is likely a compensatory mechanism induced to degrade the accumulating acetylcholine and thus reestablish the neurons normal activity. We have included this explanation in the Discussion section (L563-565): “Hence, induction of AChE activity could be a compensatory response mechanism to degrade excess acetylcholine, as has been suggested in zebrafish exposed to 0.87 to 3.51 mg/L sulfoxaflor for 24 to 96 hours.”

Reviewer #2: The study “Transcriptome analyses in juvenile yellow perch (Perca flavescens) exposed in vivo to clothianidin and chlorantraniliprole: possible sampling bias” examined the effects of two insecticides on their own and in combination on juvenile yellow perch liver transcriptomes and brain acetylcholinesterase activity. I would first like to commend the authors on the extent of work that went into the study, and their ability to communicate their findings in a comprehensive and succinct manner.

However, despite finding potential impacts of insecticide impacts on genes associated with calcium signaling, and changes in activity of acetylcholinesterase activity in the brain, a large extent of their significant results appear to be due to differences in sampling times for their treatment groups. I appreciate the effort the authors put into verifying potential impacts of sampling time by examining similar clock genes in rainbow trout, but it does unfortunately highlight an important error in the design of the experiment. If the authors wish to publish the study in its current form, I believe that discussion of the differentially transcribed clock genes needs to be greatly reduced. The connection the authors make between calcium signaling genes and clock genes (page 20) also raises questions regarding whether transcription of calcium signaling genes could also be sensitive to the sampling time. Considering the arguments made by the authors on lines 469-470 – “At this stage, it is therefore impossible to conclude on the transcriptional effect of CH on circadian gene regulation” and that they emphasize that these genes regulate 10-20% of the expressed genome (lines 471-472), it becomes difficult to discern what this study offers with respect to our knowledge on the effects of the studied insecticides on the yellow perch liver transcriptome.

Response: We understand the reviewer’s point and agree with the suggestion. The text from the sub-section Functional analyses of differentially transcribed genes was revised and a large part of the discussion on the differentially transcribed clock genes was deleted (e.g. L444-451, L460-495).

Minor comments:

Lines 269-276 – can the authors provide further justification as to why juvenile rainbow trout were an acceptable alternative to yellow perch? This point may be mute, given the end result. 

Response: Explanation on the reason for using rainbow trout is given on L270 - mRNA levels of circadian gene orthologs were also measured by qRT-PCR in unexposed juvenile rainbow trout (Oncorhynchus mykiss) due to the lack of YP stock availability outside of the annual breeding period. Additional information to justify the use of rainbow trout was added to the text on L279: Both fish species are teleosts and the controlled breeding of female rainbow trout in our laboratory allowed us to reproduce the exposure conditions of perch regarding size, weight, and carrying capacity. The sequencing of trout genome also facilitated the qRT-PCR analysis.

Line 307-308 – were differences in weight and length among treatment groups analyzed statistically?

Response: Yes. The results of the statistical comparison among treatment groups for fish length and weight have been added on L316-317. 

Line 314 – Can the authors include their reasoning on why a minimum of 1 CPM in at least ten samples was chosen as the threshold for DTG filtering. 

Response: It is recommended to filter out lowly expressed genes because very low counts across all samples provide little evidence for differential expression and because they add to the multiple testing burden when estimating false discovery rates, reducing the statistical power to detect differentially expressed genes. As a rule of thumb, the filtering threshold is chosen by identifying the CPM (counts per million) that corresponds to an actual count of 10 for the library sizes, which in our dataset is 1. CPMs are used rather than actual counts, as the latter does not account for differences in library sizes between samples. The CPM threshold is required in at least 10 samples because this is the sample size for the groups in our experiment. This text has been added on L323-329: Because very low counts across all samples provide little evidence for differential expression and because they add to the multiple testing burden when estimating false discovery rates, lowly expressed genes that did not reach a minimum of 1 count per millions reads (CPM) in at least ten samples were filtered out. This CPM value corresponds to a minimum of 10 counts in our librairies, which is the recommended threshold (Chen et al 2023).

Can the authors elaborate on how the comparison was made between the CLO+CH and M effects on differentially transcribed genes. Response: Contrasts analyses were done to compare CLO+CH and M effects. Contrasts are specified using the corresponding linear model formula. For instance, B-A refers to the comparison of group B to A, using A as the denominator in the comparison. FC were therefore calculated by dividing the transcription values of each condition by the common denominator A (control): CLO/A, CH/A, M/A. For the interaction, the test verifies that exposure to the mixture differs from the sum of the effects of the 2 pesticides taken individually: (M+A)/(CH+CLO).

Responses to the publisher –

Response: The manuscript has been revised based on the two template documents.

Response: Funding information has been added in the system. No grant number is associated with this funding. 

3. Thank you for stating the following in the Acknowledgments Section of your manuscript. We note that you have provided funding information that is currently declared in your Funding Statement. However, funding information should not appear in the Acknowledgments section or other areas of your manuscript. Please remove any funding-related text from the manuscript and let us know how you would like to update your Funding Statement. Please include your amended statements within your cover letter; we will change the online submission form on your behalf. 

Response: The funding information has been deleted from the Acknowledgement Section of the manuscript.

4. PLOS requires an ORCID iD for the corresponding author in Editorial Manager on papers submitted after December 6th, 2016. Please ensure that you have an ORCID iD and that it is validated in Editorial Manager. 

Response: The ORCID information has been updated in the corresponding author profile and added in the submission system.

---

## [Decision Letter · Decision Letter 1]

19 Mar 2024

PONE-D-23-30911R1Transcriptome analyses in juvenile yellow perch (Perca flavescens) exposed in vivo to clothianidin and chlorantraniliprole: possible sampling biasPLOS ONE

Dear Dr. Houde,

Thank you for submitting your manuscript to PLOS ONE. After careful consideration, we feel that it has merit but does not fully meet PLOS ONE’s publication criteria as it currently stands. Therefore, we invite you to submit a revised version of the manuscript that addresses the points raised during the review process.

We look forward to receiving your revised manuscript.

Kind regards,

Adekunle Akeem Bakare, Ph.D.

Academic Editor

PLOS ONE

Journal Requirements:

Additional Editor Comments:

The authors have addressed all the queries raised by Reviewers 1 and 2. In order to further improve the manuscript, I advise the authors to address the minor but important correction raised by Reviewer 3. Once they address this minor correction, I recommend acceptance of the manuscript.

Reviewers' comments:

Reviewer's Responses to Questions

**Comments to the Author**

1. If the authors have adequately addressed your comments raised in a previous round of review and you feel that this manuscript is now acceptable for publication, you may indicate that here to bypass the “Comments to the Author” section, enter your conflict of interest statement in the “Confidential to Editor” section, and submit your "Accept" recommendation.

Reviewer #1: All comments have been addressed

Reviewer #3: All comments have been addressed

2. Is the manuscript technically sound, and do the data support the conclusions?

Reviewer #1: (No Response)

Reviewer #3: Partly

3. Has the statistical analysis been performed appropriately and rigorously? 

Reviewer #1: (No Response)

Reviewer #3: Yes

4. Have the authors made all data underlying the findings in their manuscript fully available?

Reviewer #1: (No Response)

Reviewer #3: Yes

5. Is the manuscript presented in an intelligible fashion and written in standard English?

Reviewer #1: (No Response)

Reviewer #3: Yes

6. Review Comments to the Author

Reviewer #1: (No Response)

Reviewer #3: PONE-D-23-30911R1

The goal of the study was to evaluate the transcriptional and biochemical effects of these two pesticides on juvenile yellow perch exposed for 28d to environmental doses of each compound alone and in a mixture under laboratory/aquaria conditions. The authors used a transcriptional approach to examine the effects of two insecticides individually and in combination on juvenile yellow perch liver transcriptomes and brain acetylcholinesterase activity. The manuscript is well written and describes a comprehensive study. The results describe the potential impacts of insecticide impacts on genes associated with calcium signaling, and changes in activity of acetylcholinesterase activity in the brain,

This reviewer was not part of the original review and has been brought on board to evaluate the first revision. The foundation of the data used to elucidate the molecular mechanisms of the pesticides is based on RNA sequencing and qRTPCR. The experiments were well designed and performed and were not criticized in the first review, this reviewer has some concerns about the details of the quantitative transcriptomics experiment. These concerns are not likely to be fatal however, they should be addressed.

The authors are referred to section 7.4.1. PCR efficiency in the following document, provided with the review.

Bustin et al. 2009 MIQE Minimum Information for Publication of Quantitative Real-Time PCR. Clinical Chemistry 55:4 611–622.

7. PLOS authors have the option to publish the peer review history of their article (what does this mean?). If published, this will include your full peer review and any attached files.

Reviewer #1: No

Reviewer #3: **Yes: **Macdonald Wick

---

## [Author Response · Author response to Decision Letter 1]

22 Mar 2024

As requested by the reviewer we have added information on the efficiency of the PCR analyses. This sentence has been added on L275 - Percent efficiencies (90-110) and the coefficient of determination (r2>0.980) of calibration curves respected the standards recommended in Taylor et al (46). The reference has also been added to the list.

---

## [Editor Report · Decision Letter 2]

28 Mar 2024

Transcriptome analyses in juvenile yellow perch (Perca flavescens) exposed in vivo to clothianidin and chlorantraniliprole: possible sampling bias

PONE-D-23-30911R2

Dear Dr. Houde,

We’re pleased to inform you that your manuscript has been judged scientifically suitable for publication and will be formally accepted for publication once it meets all outstanding technical requirements.

Kind regards,

Professor Adekunle A. Bakare, Ph.D.

Academic Editor

PLOS ONE

Additional Editor Comments (optional):

The authors have attended to all the queries. I recommend acceptance of this article for publication.
---

## [Editor Report · Acceptance letter]

5 Apr 2024

PONE-D-23-30911R2 

PLOS ONE

Dear Dr. Houde, 

I'm pleased to inform you that your manuscript has been deemed suitable for publication in PLOS ONE. Congratulations! Your manuscript is now being handed over to our production team.

Kind regards, 

on behalf of

Professor Adekunle Akeem Bakare 

Academic Editor

PLOS ONE